# Secondary bone marrow graft loss after third-party virus-specific T cell infusion: Case report of a rare complication

Michael D. Keller [1,2,3,10], Stefan A. Schattgen [4,10],
Shanmuganathan Chandrakasan [5], E. Kaitlynn Allen [4],
Mariah A. Jensen-Wachspress[1], Christopher A. Lazarski [1], Muna Qayed[5],
Haili Lang[1], Patrick J. Hanley[1,3,6], Jay Tanna[1,6], Sung-Yun Pai [7], Suhag Parikh [5],
Seth I. Berger [8], Stephen Gottschalk [4], Michael A. Pulsipher[9,11],
Paul G. Thomas [4,11] & Catherine M. Bollard [1,3,6,11] ✉

Virus-specific T cells (VST) from partially-HLA matched donors have been effective for treatment of refractory viral infections in immunocompromised patients in prior studies with a good safety profile, but rare adverse events have been described. Here we describe a unique and severe adverse event of VST therapy in an infant with severe combined immunodeficiency, who receives, as part of a clinical trial (NCT03475212), third party VSTs for treating cytomegalovirus viremia following bone marrow transplantation. At one-month post-VST infusion, rejection of graft and reversal of chimerism is observed, as is an expansion of T cells exclusively from the VST donor. Single-cell gene expression and T cell receptor profiling demonstrate a narrow repertoire of predominantly activated CD4$^+$ T cells in the recipient at the time of rejection, with the repertoire overlapping more with that of peripheral blood from VST donor than the infused VST product. This case thus demonstrates a rare but serious side effect of VST therapy.

Viral infections are a common cause of recurrent and potentially life-threatening infections in immunocompromised patients, including patients with inborn errors of immunity (IEI) and in those undergoing bone marrow transplant (BMT)[1,2]. In patients with severe combined immunodeficiency (SCID), the presence of active infections at the time of BMT has been shown to worsen survival[3,4]. Common infections complicating transplantation for patients with SCID include cytomegalovirus and respiratory viruses such as adenovirus and RSV; unfortunately, antiviral therapies in this setting rarely enable clearance prior to immune reconstitution[5]. Advances in BMT methodologies, including ex vivo and in vivo T cell depletion have expanded donor options for IEI patients, but also result in delayed T cell reconstitution and therefore, delayed viral immunity[6,7]. Primary and secondary graft failure are also potential risks, particularly in the setting of graft manipulation and reduced intensity conditioning[8].

Given the critical role of T cell immunity in controlling viral infections, adoptive immunotherapy with allogeneic virus-specific T cells (VSTs) has been utilized in many Phase I-II trials with antiviral efficacy

[1]Center for Cancer and Immunology Research, Children's National Hospital, Washington, DC, USA. [2]Division of Allergy and Immunology, Children's National Hospital, Washington, DC, USA. [3]GW Cancer Center, George Washington University, Washington, DC, USA. [4]Department of Immunology, St Jude Children's Research Hospital, Memphis, TN, USA. [5]Aflac Cancer and Blood Disorders Center, Children's Hospital of Atlanta, Atlanta, GA, USA. [6]Division of Blood and Marrow Transplantation, Children's National Hospital, Washington, DC, USA. [7]Center for Cancer Research, National Cancer Institute, Bethesda, MD, USA. [8]Center for Genetic Medicine Research, Children's National Hospital, Washington, DC, USA. [9]Division of Pediatric Hematology/Oncology, Huntsman Cancer Institute, University of Utah, Salt Lake City, UT, USA. [10]These authors contributed equally: Michael D. Keller, Stefan A. Schattgen. [11]These authors jointly supervised this work: Michael A. Pulsipher, Paul G. Thomas, Catherine M. Bollard. ✉e-mail: cbollard@childrensnational.org

tied to the expansion of VSTs in vivo[9–11]. VSTs derived from either BMT donors or healthy, third-party donors have been infused into patients with active virus infection after BMT with low incidence of graft-versus-host-disease[12]. Moreover, other toxicities, including cytokine release syndrome rarely been reported following VST therapy in this setting[13,14].

We describe a child with RAG1 SCID who experiences secondary graft failure after haploidentical transplantation in which third-party VSTs, are administered for treatment of CMV reactivation, and show significant expansion comprising 100% of the T-cell compartment at the time of graft rejection. In this report, we show that this event correlates with the presence of activated T cells that were extremely rare in infused VST product, and occurs in the absence of CMV activity, suggesting that this is an off-target event due to rare lymphocytes that were not virus-targeting. This case highlights the rare risk of graft rejection following third-party VST infusion, and highlights the need for further studies of safeguards to prevent alloimmune reactions following immune effector cell therapies.

## Results

### Patient Clinical Description

The patient was a female infant diagnosed with RAG1 SCID (homozygous p.Thr403Ile) at 2 weeks following abnormal newborn screen result. CMV viremia was detected at 1 month of age, and treated with ganciclovir and Cytogam, followed by valganciclovir prophylaxis. The patient received an αβTCR/CD19-depleted **paternal** donor haploidentical bone marrow transplant (haploBMT) at 3.5 months of age following pre-conditioning with fludarabine, PK-adjusted busulfan, rATG, and thiotepa. Neutrophils engrafted on day +16, and platelets on day +13. Chimerism analysis on day +21 showed 96% myeloid and 92% donor T-cell chimerism. CMV viremia re-emerged on day +6 after haploBMT, leading to the initiation of foscarnet treatment and cytogam starting on day +13. Ganciclovir was introduced on day +22 due to the increasing viral load, but it was switched back to foscarnet on day +28 after CMV resistance testing revealed a UL97 mutation (C607F). Cidofovir was added to the treatment regimen on day +33 (Fig. 1A). There were no signs of CMV disease, including no hepatitis or colitis. She had bilateral interstitial densities in her lungs, but she experienced no respiratory distress or oxygen requirement. The family was consented on a clinical trial (NCT03475212), and at 42 days post-BMT, the infant received a $2 \times 10^7$ cells/m² dose of third-party VSTs derived from a female donor who was matched at 2 HLA alleles (HLA-A68:01 and HLA-DPB1*04:01) with the BMT donor (male) and a single allele (HLA-A68:01) with the patient. Due to persistent CMV viremia, ganciclovir was re-initiated on day +63. On day +68 (28 days post-VST infusion), the patient developed fever and anemia, followed by erythroderma, hepatitis, and pancytopenia. She exhibited hepatomegaly with

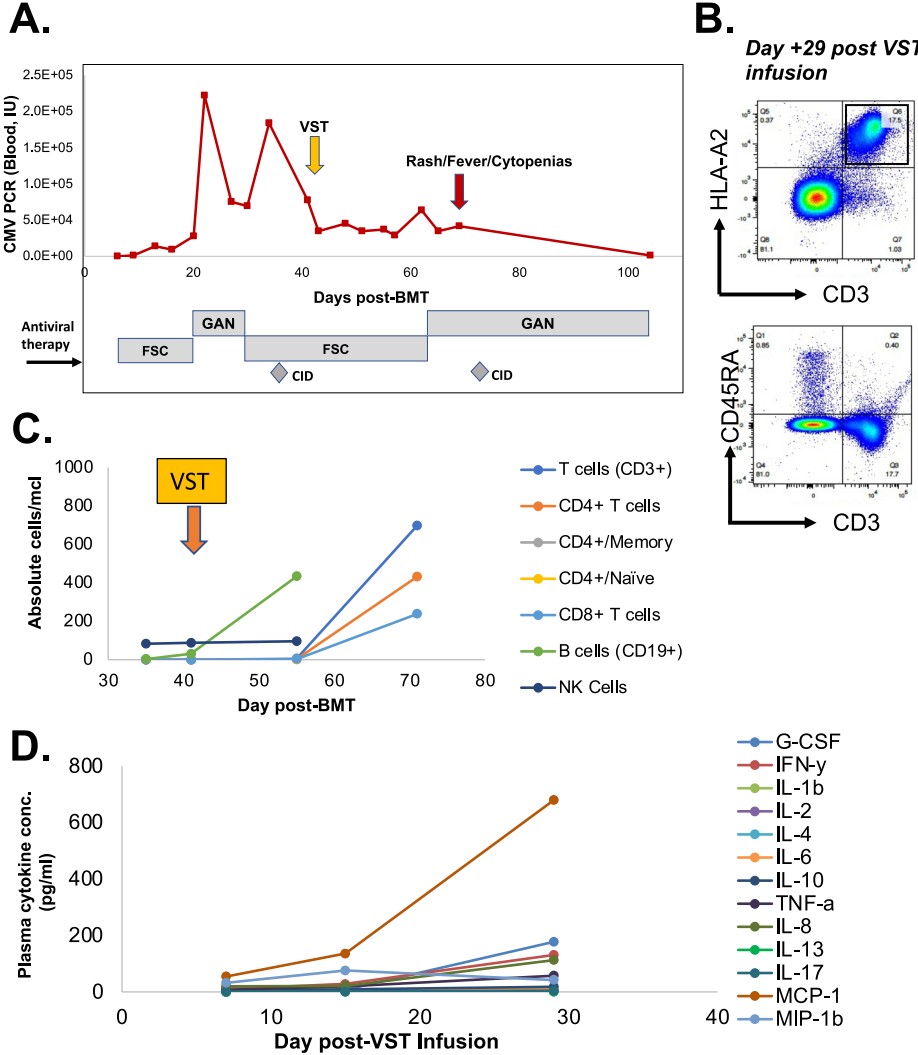

**Fig. 1 | Viral and Immunologic Patient Data. A** CMV viral loads and antiviral medications over time post-BMT and relative to VST infusion (orange arrow) and onset of clinical symptoms at onset of secondary rejection (red arrow). GAN=ganciclovir; FSC=foscarnet; CID=cidofovir. **B** T cell phenotype from recipient at time of rejection (day +72 post-BMT, +29 post-VST infusion). **C** Lymphocyte counts over time relative to BMT and VST infusion (orange arrow). **D** Plasma cytokines relative to VST infusion.

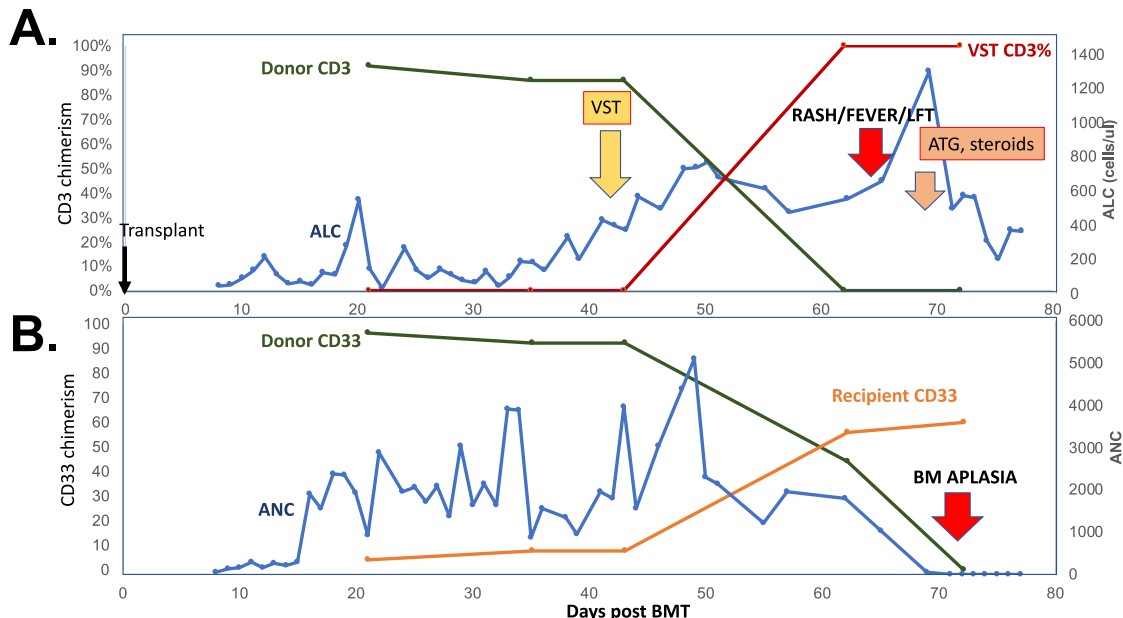

**Fig. 2 | Longitudinal Chimerism and Leukocyte counts. A** CD3 chimerism in BMT donor (green line) and VST donor (red line), absolute lymphocyte count (blue line) versus day post-BMT. VST infusion on day +42 is (yellow arrow), onset of symptoms at time of rejection (red arrow), and treatment with anti-thymoglobulin and steroids (orange arrow) are displayed. **B** CD33 chimerism in BMT donor (green line) and recipient (orange line) and absolute neutrophil count (blue line) versus day post-BMT. diagnosis of bone marrow aplasia (lower red arrow).

elevated AST and ALT levels at 305 (23–83 U/L) and 435 (6–50 U/L), respectively, but maintained normal synthetic function. Peripheral flow cytometry similarly showed a rise in effector memory T cells expressing the HLA-A02 antigen (Fig. 1B-C). Plasma cytokines demonstrated marked elevation of IFN-γ, IL-8, G-CSF, and MCP-1 at day +71 (Fig. 1D). Bone marrow biopsy on day +72 showed a profoundly hypoplastic marrow, and chimerism testing from marrow and blood showed loss of donor CD3 and CD33 chimerism, and CD3 chimerism exclusively matching the VST donor (Fig. 2). Evaluation of peripheral T cells showed no activity against CMV antigens (Fig. 3A). The patient was treated with hATG and corticosteroids on day +73 post-BMT (day 31 post VST infusion) and proceeded to a ***maternal*** donor haploBMT on day +87 after preconditioning with ATG, thiotepa, fludarabine, Cytoxan, and TBI. However, she developed hepatic veno-occlusive disease and progressive respiratory disease thereafter with ongoing cytopenias. She died from bacteremia in the setting of pancytopenia on day +104 post-BMT (day +62 post-VST infusion).

## Donor and VST product evaluations

The VST product utilized in this case was generated using a 10-day expansion protocol from peripheral blood leukocytes and targeted CMV, EBV, and adenovirus. The donor was a 46-year-old G3P3 woman with a history of allergies, but was otherwise healthy. Standard donor health screening and infectious disease testing were negative. The product met all established release criteria and was predominantly αβ T cells with both CD4 and CD8+ populations with minimal presence of other lymphocyte populations (Fig. 3B). Specificity mapping of the product showed CMV-IE1 targeting restricted by HLA-A68:01 (Fig. 3C), which was shared with the recipient (Supplementary Table 1).

Due to the gender and HLA mismatch between the VST and BMT donors, we tested the VST product (P0230D) for general alloreactivity as well as specificity for described or novel alloantigens. Mixed lymphocyte cultures of the VST product with HLA-mismatched PBMCs showed modest T cell proliferation at day 5, whereas mixed lymphocyte culture of the VST product with irradiated cells from the paternal marrow product yielded no T cell proliferation beyond background at day 5 (Supplementary Fig. 1, HLA details in Supplementary Table 1).

Pentamer staining of the VST product with described HLA-A02-restricted Y chromosome (H-Y) epitopes[15–23] were negative (Supplementary Fig. 2A). Testing of the product for T cell reactivity against peptide pools encompassing known H-Y antigens (SMYC, RPS4Y, UTY, USP9Y, JARID1D, PCDH11Y) as well as described variants in these antigens via IFN-γ ELISpot (Supplementary Fig. 2B) was also negative. In order to investigate new potential alloantigens, we performed whole genome sequencing on the VST donor, both sons of the VST donor, and the first (male) BMT donor. We evaluated for missense or non-frameshift indel variants that were shared by the BMT donor and either of the sons, but absent in the VST donor, which would therefore serve as possible neoantigens. Twenty variants were identified that fit this pattern (Supplementary Table 2), and a pool of overlapping 15-mer peptides was produced to test these possible neoantigens. PBMCs from the VST donor were expanded against pools for RPS4Y, SMYC, or the neoantigens, and testing at day 10 for reactivity against these pools by intracellular cytokine staining showed trace reactivity for the three pools based on IFN-γ and TNF−α expression (Supplementary Fig. 3).

### *Previous use of the VST Product was without complication*

The same VST product had been utilized to treat a 15-year-old female (P0264) who had undergone umbilical cord blood transplantation from a female donor for treatment of sickle cell anemia and had CMV reactivation with CMV retinitis. She received three doses of third-party VST infusions at $2 \times 10^7/m^2$/dose on days +401, 436, and 541 post-transplant, of which the second infusion was from donor P0230D. The VST product derived from donor P0230D was HLA matched at 3 of 10 antigens at low resolution, with high-resolution match at HLA-A68:01, which mediated anti-CMV IE1 activity (Supplementary Table 3). Patient P0264 had no adverse reactions following infusion of the VST product derived from donor P0230D, and at the day 45 follow-up, the patient had stable retinal disease.

### *Single-cell gene expression and TCR studies showed limited clonotype diversity* in vivo

In order to determine the phenotype and origin of T cells present in the recipient at the time of rejection, single-cell gene expression (GEX)

with TCR profiling and bulk TCR repertoire sequencing were performed on PBMCs from the patient, cells from the VST product, and PBMCs from the VST donor prior to product expansion. Comparison of these samples showed a limited number of clonotypes in common between the recipient and the VST product, with 52 TCRα and 8 TCRβ shared between the VST product and the recipient (Fig. 4). Bulk TCRB sequencing showed only 5 clonotypes in common with the VST product and 7 in common with PBMCs from the VST donor (3 detected in both), and none of these clonotypes were seen at high frequency in the recipient compared with the VST donor and product. Single-cell GEX showed that the majority of the T cells detected in the recipient's peripheral blood were activated CD4 T cells (Fig. 5). There was no single dominant clonotype detected, as demonstrated by clonotype frequencies in the recipient as well as the VST product and PBMCs from the VST donor (Supplementary Fig. 4A-B). T cell clonotype diversity was predictably lower in the VST product and recipient compared with the VST donor PBMC (Supplementary Fig. 4C-D). Comparison of the overlapping clonotypes identified in the VST donor and recipient to public TCR databases (VDJdb, MCPAS[24,25]) yielded a small number of public clonotypes, with numerous clonotypes tied to epitopes from Yellow Fever Virus (Supplementary Fig. 5). When evaluating all clonotypes in the recipient at day +30, clonotypes tied to influenza A and HTLV-1 were also identified.

TCR profiling was performed on the sorted T-cells from the VST donor that reacted to alloantigens RPS4Y, SMYC, or the neoantigen pool by IFN-γ capture assay following T-cell restimulation with peptide pools. The sorted cells were matched to the bulk and single-cell TCR sequences from the pre/post expansion VST product and the post-infusion samples above. Only four TCRα and TCRβ single chain sequences combined from the alloantigen-specific cells matched the recipient bulk TCR sampling, and four paired TCRs were matched to

TCR sequences in the 10X dataset at day +28 (Supplementary Fig. 6A). The largest clone was a mucosal-associated invariant T (MAIT) cell and the remaining paired clones were CD4+ effectors marked by expression of *ITGAL* (CD11a), *TNFRSF18* (GITR), and *TNFRSF4* (OX40) (Supplementary Fig. 6B).

## Discussion

In this report, we describe the occurrence of secondary graft rejection after immune effector cell therapy in an infant with RAG1-SCID. Secondary graft rejection is well described after BMT, and recent studies of αβTCR/CD19 depleted BMT procedures have described variable graft failure rates of between 3 and 16%. Though intrinsic graft failure cannot be completely ruled out, the striking expansion of T cells from the VST donor makes it likely that they played a role in the occurrence of graft rejection in this case. Previously reported third-party VST therapy studies have described low levels of GVHD and rare (<2%) cases of cytokine release syndrome. However, third-party VST-mediated graft failure has never previously been reported. In this case, persisting T cells in the recipient were shown to not be CMV-specific in spite of CMV specificity in the infused product, and in fact the persisting clonotypes detectable in the patient at the time of graft rejection were exceedingly rare in the product. Notably, more clonotypic overlap was detected between the recipient and the peripheral blood sample from the donor compared with the VST product. In spite of extensive evaluations, no single alloantigen was identified in this case, and no single clonotype was predominant in the recipient at the time of rejection, eliminating the possibility of an individual clone that mediated rejection.

The underlying diagnosis of this child coupled with the use of ex vivo T cell-depleted transplant without post-transplant immunosuppression resulted in profound lymphopenia that presumably

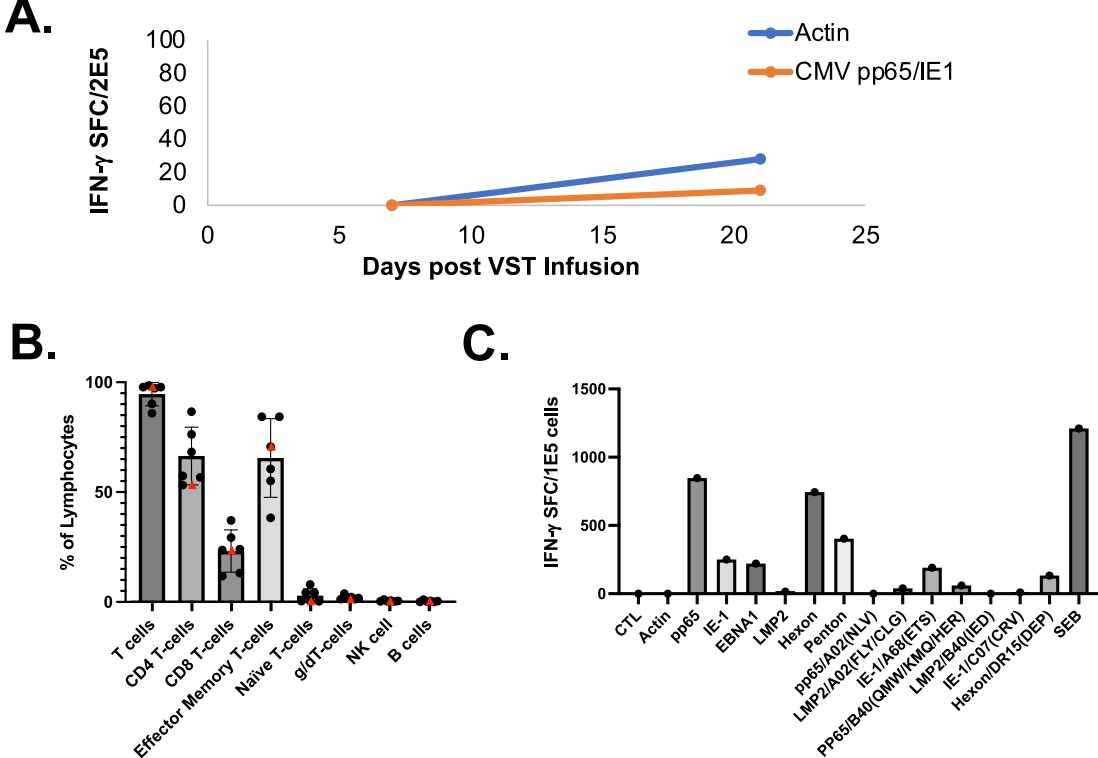

**Fig. 3 | Extended Immunologic Studies of the Recipient and VST Product.**
**A** Evaluation of CMV specificity in recipient peripheral blood leukocytes by interferon-γ ELISpot testing following VST infusion. SFC = spot forming colonies. **B** Phenotype of infused VST product (shown as a red triangle) as well as other control VST products (*n* = 5). Bar=mean, whiskers: standard deviation, red triangle: median. **C** Antiviral specificity of infused VST product to whole viral antigens and specific HLA-restricted viral peptides by interferon-gamma ELISpot.

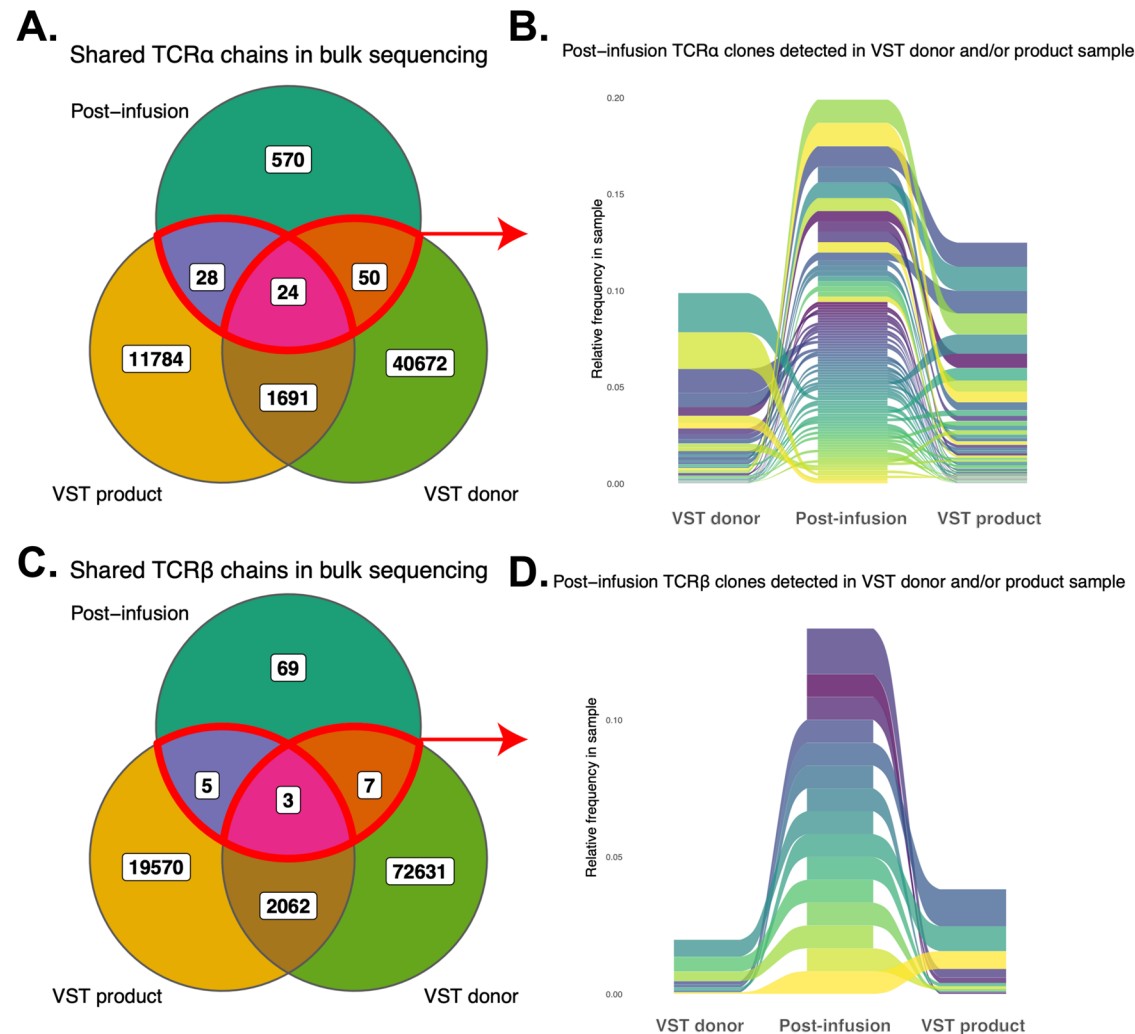

**Fig. 4 | T cell receptor clonotype overlap between recipient, virus-specific T cell product, and VST donor peripheral blood. A** Venn diagram of overlapping TCRα clonotypes at day +30 post-VST infusion. **B** Ribbon diagram of TCRα clonotypes between recipient, VST product, and VST donor. **C** Venn diagram of overlapping TCRβ clonotypes at day +30 post-VST infusion. **D** Ribbon diagram of TCRβ clonotypes between recipient, VST product, and VST donor.

facilitated engraftment and dramatic expansion of rare T cell clones in vivo following VST infusion. Why these clones persisted over the CMV-specific clones which made up a much larger proportion of the VST product is unknown. Production of VST products using a 10-12-day ex vivo expansion protocol has been demonstrated to result in a limited number of high-frequency T cell clones. This report suggests that potentially alloreactive clones, while rare, may still exist in these final products at extremely low frequencies. This may only be of concern when, as in this case, they are infused into a profoundly T-cell-depleted patient early post-transplant. In studies to date, third-party VST infusions have not resulted in any known graft rejection adverse events, likely because these rare cells are heavily out-competed by the virus-specific populations and typically, patients have enough engrafted T-cell function to facilitate rejection of the third-party VST products within 3 months postinfusion. However, donor factors that may result in allosensitization or immune-mediated disease, including prior pregnancies, transfusions, or prior autoimmune disease, may also impact the risk of rare but serious complications after immune effector cell therapy. In response to this occurrence, we instituted revised suitability criteria for VST donors (Supplementary Table 4) and also modified recipient eligibility (Supplementary Table 5) to address theoretical risk factors that may impact the safety of third-party VSTs post allogeneic bone

marrow transplant. These require further study to clarify the true risk factors for alloreactions.

In summary, we highlight a rare but serious risk of third-party VST therapy in a patient with RAG1-SCID who experienced secondary graft rejection. Though unique, this case highlights the need for further studies of patient and donor factors that may influence the safety of immune effector cell therapies.

## Methods

### Patients
The patient was treated under a protocol approved at institutional review boards of participating centers as well as the drug safety monitoring board of the Pediatric Transplantation and Cellular Therapy Consortium (PTCTC) and participating centers (Children's National Hospital, Children's Hospital of Los Angeles, Children's Hosital of Atlanta, and St Jude's Children's Research Hospital). Blood samples, clinical information, and permission for publication were obtained from participants under informed consent approved by Institutional Review Boards at each institution in accordance with the Declaration of Helsinki.

### Whole genome sequencing and variant filtering
Genomic DNA was extracted and processed using the TruSeq DNA PCR-Free whole genome library preparation kit (Illumina,

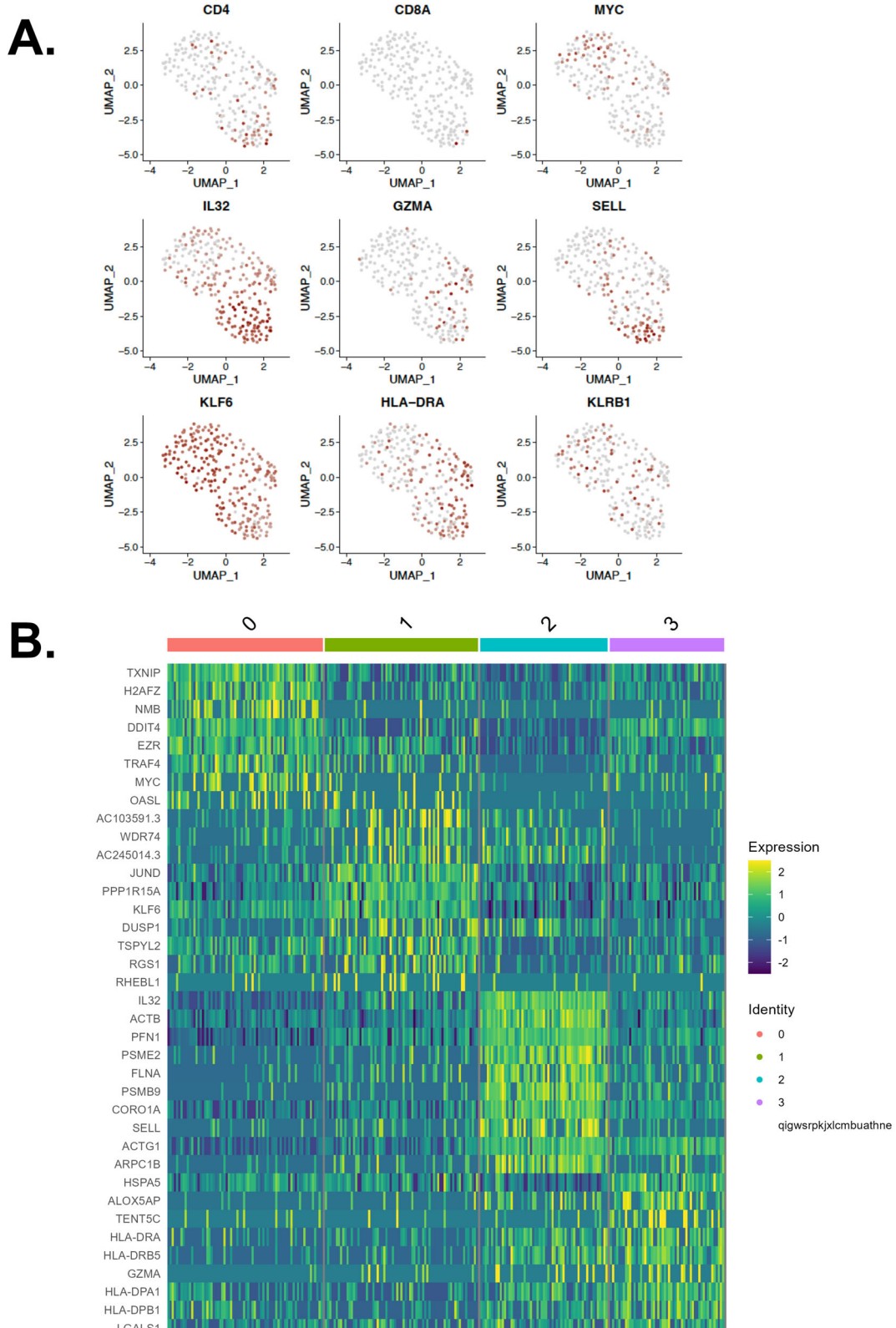

**Fig. 5 | Single-cell transcriptomic profile of recipient T cells at time of rejection. A** Selected principal component analyses for T cell phenotype and activation genes in recipient cells. **B** Heat map of gene expression in four clonotypes present in the recipient at day +30 post-VST infusion.

Cat#20015962). These regions were sequenced by massive parallel (NextGen) sequencing with 150 bp paired-end reads on the Illumina NovaSeq 6000 Sequencing System. DNA sequencing reads were aligned to human genome build UCSC hg19 (hs37d5) in BaseSpace Sequence Hub using DRAGEN Germline Pipeline (v3.4.5). Variant calls

files were annotated using annovar (v2020-06-08) with refseq gene function annotations and gnomad allele frequency (population max). Annotated vcf files were filtered using bcftools (v1.12) to include variants missense and inframe exonic indel variants (potential neoantigens), with an allele frequency below than 10% in gnomad (rare enough

to be consistent with event frequency), present in the in the bone marrow donor (potential neoantigen present), absent in the proband's mother (potential for antigen sensitization), present in at least one sibling (potential source for maternal sensitization), and with genotype quality (GQ) greater than 15 for all samples.

## Generation of antigen-specific T cells

Expansion of alloantigen-specific T cells from PBMCs was performed using a rapid expansion protocol as previously described[26]. Briefly, PBMCs were pulsed with a mix of overlapping peptide pools encompassing the proteins of interest (1ug/antigen/15 × 10$^6$ PBMCs) for 30 minutes at 37 °C. Peptide libraries of 15-mers with 11 amino acid overlaps encompassing previously described alloantigens, variant alloantigens, and candidate neoantigens in were generated (A&A peptide, San Diego, CA, USA) from published reference sequences, and were pooled equally by mass and reconstituted to a working concentration of 1ug/µl. All utilized sequences are listed in Supplementary Data 1-2. After incubation, cells were resuspended with IL-4 (400 IU/ml; R&D Systems Cat#BT-004, Minneapolis, MN) and IL-7 (10 ng/ml; R&D Systems Cat#BT-007) in CTL media consisting of 45% RPMI (GE Healthcare, Logan, UT), 45% Click's medium (Irvine Scientific, Santa Ana, CA), 10% fetal bovine serum, and supplemented with 2 mM GlutaMax (Gibco, Grand Island, NY). Cytokines were replenished on day 7. On day 10, cells were harvested and evaluated for antigen specificity and functionality.

## IFN-γ enzyme-linked immunospot (ELISpot) assay

Antigen specificity of T cells was measured by IFN-γ ELISpot (Millipore, Cat#MSHAS4510, Burlington, MA). T cells were plated at 1 × 10$^5$/well with no peptide, actin (JPT, Cat#PM-ACTS, Berlin DE), or each of the individual experimental peptide pools (1ug/peptide/well). Plates were sent for IFN-γ spots forming colony (SFC) counting (Zellnet Consulting, Fort Lee, NJ, USA).

## Flow cytometry

VSTs were stained with fluorophore-conjugated antibodies against CD4, CD8, TCRαβ, TCRγδ, CD16, CD19, and CD56 (Miltenyi Biotec, Bergisch Gladbach, Germany; BioLegend). All samples were acquired on a CytoFLEX cytometer (Beckman Coulter, Brea, CA). Intracellular cytokine staining was performed as follows: 1 × 10$^6$ VSTs were plated in a 96-well plate and stimulated with pooled pepmixes or individual peptides (200 ng/peptide/well) or actin (control) in the presence of brefeldin A (Golgiplug; BD Biosciences, Cat#BD555029, San Jose, CA) and CD28/CD49d antibodies (BD Biosciences, Cat#347690) for 6 hours. T-cells were fixed, permeabilized with Cytofix/Cytoperm solution (BD Biosciences) and stained with IFN-γ and TNF-α antibodies (Miltenyi Biotec). Pentamer staining was performed using APC-conjugated pentamers (Proimmune, Oxford, UK) per manufacturer's guidelines. Complete antibody panels and dilutions, as well as all utilized pentamers are listed in Supplementary Tables 6-12. All gating strategies are listed in Supplementary Fig. 7. Data was analyzed with FlowJo X (FlowJo LLC, Ashland, OR, USA).

## Chimerism testing

Assessment of DNA chimerism was performed using the GlobalFiler™ PCR Amplification Kit (Applied Biosystems, Cat#4476135, Waltham, MA, USA). The GlobalFiler™ kit utilizes 21 autosomal STR markers (D3S1358, vWA, D16S539, CSF1PO, TPOX, D8S1179, D21S11, D18S51, D2S441, D19S433, TH01, FGA, D22S1045, D5S818, D13S317, D7S820, SE33, D10S1248, D1S1656, D12S391, D2S1338) plus Amelogenin as a sex determining marker with DYS391 and Y-indel as Y-chromosome markers. For each STR marker, separate PCR reactions were carried out with both positive and negative controls. PCR amplification was performed on the VERITI, 96 well thermal cycler (Applied Biosystems, USA) following the manufacturer's recommended protocols.

Assessment of the STR fragments was performed on the ABI 3500XL Genetic Analyzer (Applied Biosystems, USA) following the manufacturer's instructions. Final chimerism determinations were performed using ChimeRMarker™ Genetic Analysis software (SoftGenetics, State College, PA. USA). CD3 and CD33 subsets were isolated from whole blood via magnetic beads using the EasySep™ cell isolation kit (Cat#18981RF and #17885RF) and the RoboSep™ automated cell separator system (StemCell Technologies, Cambridge, MA.). Cell purity was assessed by flow cytometry and was >95%.

## Alloreactivity assay

T cells were utilized as effectors and either PBMCs or bone marrow donor apheresis product cells as targets for mixed lymphocyte reactions. Effector T cells were washed once with 1x PBS and spun at 450 × $g$ for 5 minutes to pellet lymphocytes. T cells were re-suspended to a final concentration of 1 × 10$^6$ cells per mL in 1 × PBS, and 5 × 10$^6$ cells were added to 5 µL of 5 µM of Cell Trace Violet for a final concentration of 5 µM in 15 mL collection tubes. T cells were incubated at 37 °C for 20 minutes, after which 10 mL of complete media was added to the cell suspension and subsequently incubated at 37 °C for additional 5 minutes. T cells were spun down and washed 1x with 10 mL of complete media, then re-suspended in 5 mL of complete media for a final concentration of 1 × 10$^6$ cells per mL for subsequent co-culture. 100 µL cell suspension was added to individual wells of a 96 well U-bottom plate for a final concentration of 1 × 10$^5$ effectors per well.

For targets, 5 × 10$^6$ PBMC or cryopreserved apheresis product cells were added to 5 µL of 5 µM Cell Trace Red for a final concentration of 1 µM in 15 mL collection tubes (Supplementary Table 13). Target cells were incubated at 37 °C for 20 minutes, after which 10 mL of complete media was added to the cell suspension and incubated at 37 °C for an additional 5 minutes. Target cells were spun down and washed 1 × with 10 mL of complete media. Labeled target cells were re-suspended in 5 mL of complete media for a final concentration of 1 × 10$^6$ of the mixed cells per mL, which were then irradiated at 25 Gy.

For mixed lymphocyte cultures, 100 µL effector cell suspension (1 × 10$^5$ effectors) and 100 µL of target cell suspension (1 × 10$^5$ targets) were added to individual wells of a 96-well U-bottom plate for a 1:1 ratio of effectors to targets. Control wells consisted of 100 µL effector cell suspension with 100 µL of media alone, 100 µL effector cell suspension with 100 µL of media plus PHA or IL2, and 100 µL target cell suspension with 100 µL of media alone. Cells were cultured at 37 °C for five days.

## Multiplex cytokine assay

Plasma samples were evaluated using the Bio-plex Pro Human 17-plex Cytokine Assay kit (Bio-Rad, Cat#M5000031YV, Hercules, CA, USA), and read on a MAGPIX system (Luminex, Austin, TX, USA).

## Flow sorting of antigen-specific T cells

To isolate antigen-specific T cells prior to single cell RNAseq and TCR sequencing, an IFN-γ capture assay (Miltenyi, Cat#130-090-433, San Diego, CA, USA) was used following re-stimulation with 15-mer peptide libraries encompassing alloantigens or neoantigens of interest. IFNγ$^+$ T cells were captured on a CytoFlex SRT Cell Sorter (Beckman Coulter, Brea, CA, USA).

## Single-cell gene expression and TCR profiling

Sorted live, CD3 + T cells from the VST donor, VST product, and post-infusion samples were processed using the V(D)J + 5' Gene Expression profiling kit (v1) from 10X Genomics (Cat# 1000265) and libraries prepared following the manufacturer's protocol. Final gene expression and TCR libraries were indexed for Illumina sequencing with the Chromium i7 multiplex kit (10X Genomics, Cat#PN-120262). Both gene expression and TCR libraries were sequenced on a NovaSeq 6000 (Illumina) with 100 bp PERs and 150 bp PERs, respectively. All TCR sequences are enclosed in Supplementary Data 3.

## 10X data processing and analysis

Individually sequenced gene expression libraries were aggregated using CellRanger (v3.1.0) with default parameters. The aggregate counts matrix was subsequently analysed using the Seurat package (v 4.1.0)[27,28] for R (v 4.1.2). Cell cycle genes were scored and regressed for during processing. TCR and genes were excluded during variable gene selection prior to principal component analysis and further dimensionality reduction and clustering. The filtered_contig_annotations.csv file for each sample was processed using the make_10x_clones_file routine in the CoNGA package[29] to generate paired TCR clonotypes.

## Single-cell and bulk TCR library prep and sequencing

Single-cell TCR amplification and sequencing after sorting into 384-well plates was performed as previously described[30]. For bulk TCR sequencing, TCRα and TCRβ chains were amplified using a 5' Rapid Amplification of cDNA Ends (RACE) with unique molecular identifiers (UMIs) for error correction essentially as described[31]. RNA was extracted using the RNeasy Micro Kit (Qiagen, Cat#74004). Reverse transcription was carried out using SmartScribe RT (Takara, Cat#639538), and Q5 polymerase (New England Biolabs, Cat#M0491S) was used during first and second round amplification. Barcoded TCRα and TCRβ amplicons generated by the second round PCR were pooled by equal volume, prepped, and indexed for sequencing on Illumina platforms using a KAPA HyperPrep Kit (Roche, Cat#KK8501). 150 bp paired-end sequencing was performed on an Illumina NovaSeq6000 by the St Jude Hartwell Center.

## Processing bulk TCR sequencing

Demultiplexing and contig assembly of paired-end FASTQ reads was done with migec (v1.2.9)[32] using the CheckoutBatch and AssembleBatch commands, respectively. VDJ junction mapping and clonotype assembly and annotation using the assembled contigs from migec was done with mixcr (v3.0.13)[33] using the analyze amplicon routine. The vdjtools v1.2.1[34] FilterNonFunctional, Correct, and Decontaminate functions were used on the filtered clonotype table outputs from mixcr for additional quality control to remove erroneous clonotypes due to PCR errors and cross-contaminating sequences between samples. Immunarch (v0.6.6) package[35] for R was used to measure sample diversity, clonality, and overlap.

## Reporting summary

Further information on research design is available in the Nature Portfolio Reporting Summary linked to this article.

## Data availability

All data are included in the Supplementary Information or available from the authors upon reasonable requests, as are unique reagents used in this Article. The raw numbers for charts and graphs, including de-identified patient data, are available in the Source Data file whenever possible. The study protocol is enclosed in Supplementary material for *NCOMMS-23-36160A*. Flow cytometry and single cell sequencing data sets are available on Zenodo [https://doi.org/10.5281/zenodo.10028505] and GenBank (Bioproject PRJNA1051284). Source data are provided with this paper.

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

## Acknowledgements

The authors would like to thank the staffs of the Pediatric Transplantation and Cell Therapy Consortium and the Center for Cancer and Immunology Research at Children's National, and Dr. Robert A Bray, Ph.D. and Dr. Howard Gebel, Ph.D for enabling this work. This work was supported by grants from the California Institute for Regenerative Medicine (to MP and MDK). This work was supported in part by funding from the Intramural Research Program, National Institutes of Health, National Cancer Institute, Center for Cancer Research.

## Author contributions

M.D.K., S.S., E.K.A., C.L., S.B., P.T., M.A.P. and C.M.B. conceived and designed the experiments; S.S., W.K.A., M.J.W., C.L., J.T., P.H., M.Q., H.L. conducted the research; M.D.K., S.S., E.K.A., M.J.W., C.L., S.Y.P., P.T., and C.M.B. analyzed data; M.D.K., S.S., S.C., E.K.A., C.L., M.Q., P.H., S.Y.P., S.P., S.B., S.G., M.A.P., P.T., and C.M.B. wrote the manuscript. All authors have read and approved the final manuscript.

## Competing interests

CMB is on the scientific advisory boards for Catamaran Bio and Mana Therapeutics with stock and/or ownership, is on the Board of Directors for Caballeta Bio with stock options and has stock in Neximmune and Repertoire Immune Medicines and serves on the DSMB of SOBI. MAP is on Advisory boards—Novartis, Gentibio, Bluebird, Vertex, Medexus, Equillium; and Study Support—Adaptive, Miltenyi. PJH is a Co-founder and Board of Directors: Mana Therapeutics; and Scientific Advisory Board: Cellevolve, Cellenkos, Capsida, MicrofluidX, Discovery Life Sciences. MDK is an author for Elsevier (Uptodate), and has received research funding from Chiesi Pharmaceuticals. SC is on SAB/ Honoraria from SOBI, honoraria from Pharming P.G.T. has consulted and/or received honoraria and travel support from Illumina, Johnson and Johnson, and 10X Genomics. P.G.T. serves on the Scientific Advisory Board of Immunoscape and Cytoagents. The remaining authors declare no competing interests.
