## [Peer Review File · Nature Communications]

Secondary bone marrow graft loss after third-party virus-specific T cell infusion: Case report of of a rare complicationREVIEWER COMMENTS

Reviewer #1 (Adoptive cell transfer therapy, viral immunity, transplantation) (Remarks to the Author):

This manuscript is a case report of graft loss after infusion of VST - a serious outcome infusion that has not been previously reported in this context. A temporal association between graft failure and in vivo expansion of a polyclonal population of T cells from the third party VSTs used to treat refractory CMV viremia is the most notable observation. The patient eventually died from transplant related complications after a second allograft from an alternate donor.

This adverse event, although rare, is important and should be published in some form.

The development of complete VST donor chimerism at the time of graft loss is striking and very unusual, but causation is not demonstrated. There are other plausible explanations – the type of transplant is known to have a high rate of graft failure as the authors note. In addition, ganciclovir was commenced a short period before graft failure and is well known to cause myelosuppression and graft loss. CMV itself can also but the low level viremia makes this a less likely cause.

Comment on the manuscript:

The clinical events are largely described clearly in the text, figures and supplemental data. It should be made clear whether or not the patient was on a VST clinical trial, what immune suppression was used and a clear timeline of the use of antiviral agents, in particular, ganciclovir close to the time of graft failure. Further detail about rash, fever, liver function derangement at the time of graft loss would be helpful – was this considered an alloreaction as seen in other haplo transplant settings, GVHD or some other etiology?

A thorough investigation of the potential for the engrafted VST clones was undertaken using several avenues of investigation.

The development of full T cell chimerism from the VST donor is reported but the method is not described. In my opinion this is the most important finding and the method should be described in detail including what method of cell sorting was used and the level of precision usually obtained.

TCR bulk sequencing was performed on the VST donor and recipient samples. More detail is needed to fully assess these results:

1. What were the time points tested? Were there recipient pre-infusion and transplant donor samples?
2. What were the basic sample metrics – total clones, clonality and diversity, proportion of top n clones (the overlapping clones occupy 10-15% of the total repertoire of the patient, what was in the remaining 85-90%? Large or small clones? Were they detectable pre-infusion and did they overlap with the transplant donor or recipient pre-VST?)

The data shown appears to show a polyclonal T cell repertoire, without dominance of an individual clone although clonality should be provided (see point 2 above). Overlap of post-infusion clones in the recipient with the VSTs was lower than expected given 100% T cell chimerism from the VST donor. Could this be due to the timing of the test being asynchronous to the 100% chimerism result? If performed at the same time as chimerism, this would support the explanation proposed in the discussion that the engrafted clones were and represent low frequency bystander clones in the VST not detected by the TCR assay.

The authors went to admirable lengths to try to identify the antigen target of the VST donor T

cells but did not identify a target. Given TCR sequencing was performed, comparison of the TCR repertoire with public databases may be of value.

VST characteristics are provided via ELISPOT which shows relative proportions of reactivity to various antigens but is uninformative about the overall proportions of VSTs to intended targets vs proportion of bystander cells. Would increased purity of the VST to the pathogen targets be of benefit?

The authors propose a set of criteria for selecting donors in the supplementary material. No scientific justification is provided for each criterion, nor is there commentary on the current selection practices and how these new proposed criteria differ. This case report does not identify any feature of the VST that can be linked to donor qualities. I do not believe it is helpful to include a set of criteria of this detail without careful explanation and justification of the evidence base. The criteria should be removed altogether and addressed in another forum. The proposal to restrict use in the early phase of highly immune suppressive transplants where VST engraftment would be more likely is reasonable but would be better supported if more detail of the recipient T cell reconstitution prior to VST were available.

Minor issues

- Typographical errors are noted

- o Line 326 "from during"

- o Supp figure 1 C – x axis labels

- Supplemental figure 5 is hard to interpret in the way it is annotated and brief figure legend – what do CB_1, CB_2, CB_3 refer to?

Reviewer #2 (Graft rejection, allo-responses) (Remarks to the Author):

This is a very interesting case report of donor bone marrow rejection by an infused third party virus-specific T (VST) cell product in an infant with primary immunodeficiency disease who developed refractory CMV viremia following the T cell-depleted donor bone marrow transplant. The paper is very interesting as a case report, as this complication of third party VST has not been previously reported. The mechanistic studies, however, do not provide clear answers and could be more incisively designed.

In particular, the main problem is that the TCR sequencing from alloreactivity assays is focused on minor histocompatibility (HY) antigens and genetic deletions producing neoantigens that would also be seen as minor histocompatibility antigens, rather than on allogeneic HLA alleles, which readily present numerous non-polymorphic peptides as alloantigens. Thus, the focus is not on the most relevant alloreactive T cells and it is not surprising that very few donor-specific minor antigen-reactive clones are detected among VST-derived T cells that expanded during the rejection and that very little reactivity against such antigens is detected. It would have been more informative to sort and sequence T cells dividing in MLRs against BM donor HLA alloantigens, which would be expected to be far more abundant among the cells that expanded in vivo and rejected the graft. If BM donor cells were not available, donor HLA allele-specific T cells could have been identified by stimulation with B-LCLs expressing particular alloantigens of the donor. Without TCR sequencing data for HLA alloantigens, the rest of the analyses are not very informative.

There are a number of problems with the data as presented:

- 1) The data in Supplemental Fig.2 are confusing for several reasons. One is that the stimulators are labelled as A02+ or A02-, but both the VST donor and the BM donor carry variants of HLA-A02 (A02:01 in the case of the BM donor and A02:11 for the VST donor). Presumably the stimulators do or do not express A02:01, but this needs to be indicated more clearly;
- 2) The allogeneic stimulators in Supplemental Figure 2 presumably have extensive HLA mismatches from the VST in addition to HLA-A02. The entire HLA of the stimulators should be shown;
- 3) It is stated in line 139 that pentamer staining with HLA-A02-restricted Y chromosome epitopes was negative, but this is not apparent from the cited figure, in which small percentages of positive cells are detected. A negative control stain is needed to determine whether these are real or not;
- 4) Line 177 states that T cells were sorted from the VST donor that reacted to Y antigens or the neoantigen pool, but the sorting method is not indicated. Was this done with tetramer/pentamer stains? The small pool of sorted positive cells seems likely to be heavily contaminated with nonspecifically stained cells. No methodology is provided for this sorting experiment;
- 5) Additional information on the TCR diversity of the VST would be useful.

RESPONSES TO REVIEWERS

Reviewer #1 (Adoptive cell transfer therapy, viral immunity, transplantation) (Remarks to the Author):

This manuscript is a case report of graft loss after infusion of VST - a serious outcome infusion that has not been previously reported in this context. A temporal association between graft failure and in vivo expansion of a polyclonal population of T cells from the third party VSTs used to treat refractory CMV viremia is the most notable observation. The patient eventually died from transplant related complications after a second allograft from an alternate donor.

This adverse event, although rare, is important and should be published in some form.

The development of complete VST donor chimerism at the time of graft loss is striking and very unusual, but causation is not demonstrated. There are other plausible explanations – the type of transplant is known to have a high rate of graft failure as the authors note. In addition, ganciclovir was commenced a short period before graft failure and is well known to cause myelosuppression and graft loss. CMV itself can also but the low level viremia makes this a less likely cause.

Comment on the manuscript:

The clinical events are largely described clearly in the text, figures and supplemental data. It should be made clear whether or not the patient was on a VST clinical trial, what immune suppression was used and a clear timeline of the use of antiviral agents, in particular, ganciclovir close to the time of graft failure. Further detail about rash, fever, liver function derangement at the time of graft loss would be helpful – was this considered an alloreaction as seen in other haplo transplant settings, GVHD or some other etiology?

RESPONSE: We appreciate the reviewer's comments and have added additional details as requested to the clinical report and figures. The patient was on a third party VST clinical trial (NCT03475212) conducted at PBMTC/PIDTC Centers throughout the USA. We have more clearly detailed what immune suppression was used and the antiviral agents used etc. Though we cannot rule out GVHD or medication-associated rejection, this rejection episode most closely resembled transfusion-associated GVHD. This assertion is especially supported by the STR data showing that 100% of the T cell population detected in the patient at the time of rejection was derived from the third-party VST donor.

A thorough investigation of the potential for the engrafted VST clones was undertaken using several avenues of investigation.

The development of full T cell chimerism from the VST donor is reported but the method is not described. In my opinion this is the most important finding and the method should be described in detail including what method of cell sorting was used and the level of precision usually obtained.

TCR bulk sequencing was performed on the VST donor and recipient samples. More detail is needed to fully assess these results:

1. What were the time points tested? Were there recipient pre-infusion and transplant donor samples?
2. What were the basic sample metrics – total clones, clonality and diversity, proportion of top n clones (the overlapping clones occupy 10-15% of the total repertoire of the

patient, what was in the remaining 85-90%? Large or small clones? Were they detectable pre-infusion and did they overlap with the transplant donor or recipient pre-VST?)

RESPONSE: We thank the reviewer for these suggestions, and we have rearranged Figure 3 in the paper to show the basic sample metrics for the bulk TCR alpha and beta chain sequencing results. The scRNAseq/TCR was performed only on the VST product, PBMCs from the VST donor (from the peripheral blood sample used to manufacture the VST product), and the recipient post-infusion at Day +30. Panels A and C show the proportion of the repertoire accounted for by different fractions of clonotypes as well as the total number of clonotypes sequenced for each sample. Panels B and D demonstrate the frequencies of the overlapping clonotypes in the VST product, donor, and recipient at day +30. TCR repertoire of the post-infusion sample obtained from the patient on day +30 post VST infusion was more diverse and less clonal than the VST infusion product, and by these measures was more like the sample from the VST donor prior to expansion. Clonotype metrics and diversity scores has been added in supplemental Figure 5.

The data shown appears to show a polyclonal T cell repertoire, without dominance of an individual clone although clonality should be provided (see point 2 above). Overlap of post-infusion clones in the recipient with the VSTs was lower than expected given 100% T cell chimerism from the VST donor. Could this be due to the timing of the test being asynchronous to the 100% chimerism result? If performed at the same time as chimerism, this would support the explanation proposed in the discussion that the engrafted clones were and represent low frequency bystander clones in the VST not detected by the TCR assay.

RESPONSE: The blood sample tested was obtained after the onset of pancytopenia, rash, and cytopenias (day +30). Therefore, any pathogenic T cells involved in rejection would likely have been present in that sample. We observed that the largest clones measured in the post-infusion sample were not necessarily the largest clones in the VST product, or in other words, extremely rare clones in the VST product could become the dominant clones in the recipient. There are also technical limitations in the TCR sequencing assay that bring our recovery to approximately 70% efficiency. Together, these data suggest that rare clonotypes in the VST donor sample were missed and therefore could not be identified in the post-infusion sample.

The authors went to admirable lengths to try to identify the antigen target of the VST donor T cells but did not identify a target. Given TCR sequencing was performed, comparison of the TCR repertoire with public databases may be of value.

RESPONSE: We thank the reviewer for this suggestion, and agree that the catalog of viral-specific TCRs has become sufficiently deep enough that matching to these could prove insightful. We performed a query of the TCR sequences observed in the patient post-infusion and compared the hits to CDR3 sequences that overlapped with the VST product versus those that were unique to the recipient. We used subset of high confidence paired TCR sequences from VDJ-db and allowed query sequences to fall within an edit distance of 1 of entry to capture highly similar sequences. Here, 20% of shared CDR3 α chains matched to entries in the reference dataset compared to 11% of sequences that were not shared with

the VST samples. Matches to TCR sequences recognizing CMV and EBV amongst others were detectable in the shared CDR3 α chains. Interestingly, while some recipient-unique sequences also recognized CMV and EBV, there were many matches to TCRs recognizing epitopes derived from Influenza and HTLV-1. These results have been added in Supplemental Figure 6

VST characteristics are provided via ELISPOT which shows relative proportions of reactivity to various antigens but is uninformative about the overall proportions of VSTs to intended targets vs proportion of bystander cells. Would increased purity of the VST to the pathogen targets be of benefit?

RESPONSE: We appreciate the reviewer's comment. The VSTs utilized in this study were produced by ex vivo expansion, as opposed to selection. Additionally, all VSTs are expanded against CMV, EBV, and adenovirus antigens, and therefore only a proportion of the final product would be specific for any given viral antigen/epitope. It is possible that T cells that are not virus-specific may remain in the final product despite a lack of antigenic stimulation during the expansion. This is indeed what we believe occurred in this case. It is possible further modification of VST production methods to purify the virus-specific component may increase the safety margin of this therapy, though further study of alloreactions are critical to identify patterns and biomarkers that correlate with risk of GVHD after T cell infusion. We have commented on this in the discussion.

The authors propose a set of criteria for selecting donors in the supplementary material. No scientific justification is provided for each criterion, nor is there commentary on the current selection practices and how these new proposed criteria differ. This case report does not identify any feature of the VST that can be linked to donor qualities. I do not believe it is helpful to include a set of criteria of this detail without careful explanation and justification of the evidence base. The criteria should be removed altogether and addressed in another forum. The proposal to restrict use in the early phase of highly immune suppressive transplants where VST engraftment would be more likely is reasonable but would be better supported if more detail of the recipient T cell reconstitution prior to VST were available.

RESPONSE: We thank the reviewer for these important comments and acknowledge that these criteria were largely driven by our regulatory interactions, and do not have full scientific evidence to justify each of these criteria. However, since these criteria were acceptable to the FDA in order for the study to be reopened, we assert, to ensure full transparency, that it is important for the BMT community to share the criteria we used. However, we are emphasizing that this is informational only as opposed to a recommendation. Moreover, per your suggestion, we have also modified the discussion to identify possible donor factors that may influence safety of adoptive therapy.

Minor issues

- Typographical errors are noted
 - o Line 326 "from during"
 - o Supp figure 1 C – x axis labels
- Supplemental figure 5 is hard to interpret in the way it is annotated and brief figure legend – what do CB_1, CB_2, CB_3 refer to?

RESPONSE: Thank you for bringing this to our attention - these have been fixed.

Reviewer #2 (Graft rejection, allo-responses) (Remarks to the Author):

This is a very interesting case report of donor bone marrow rejection by an infused third party virus-specific T (VST) cell product in an infant with primary immunodeficiency disease who developed refractory CMV viremia following the T cell-depleted donor bone marrow transplant. The paper is very interesting as a case report, as this complication of third party VST has not been previously reported. The mechanistic studies, however, do not provide clear answers and could be more incisively designed.

In particular, the main problem is that the TCR sequencing from alloreactivity assays is focused on minor histocompatibility (HY) antigens and genetic deletions producing neoantigens that would also be seen as minor histocompatibility antigens, rather than on allogeneic HLA alleles, which readily present numerous non-polymorphic peptides as alloantigens. Thus, the focus is not on the most relevant alloreactive T cells and it is not surprising that very few donor-specific minor antigen-reactive clones are detected among VST-derived T cells that expanded during the rejection and that very little reactivity against such antigens is detected. It would have been more informative to sort and sequence T cells dividing in MLRs against BM donor HLA alloantigens, which would be expected to be far more abundant among the cells that expanded in vivo and rejected the graft. If BM donor cells were not available, donor HLA allele-specific T cells could have been identified by stimulation with B-LCLs expressing particular alloantigens of the donor. Without TCR sequencing data for HLA alloantigens, the rest of the analyses are not very informative.

RESPONSE: We appreciate the reviewer's comments. In our investigations, we focused first, as the reviewer noted, on gender-related antigens, followed by neoantigens based on sequencing data, and disproved these two hypotheses. While we agree that HLA mismatch may have driven this reaction, we did not observe any notable alloreactivity of the VST product in comparison to other VST products that had been safely administered to patients without eliciting an alloreactive response in vivo. Additionally, the TCR sequencing data obtained from recipient's peripheral blood at the time of rejection did not show any single dominant clonotype, making it less likely that a single clonal population was to blame for this rejection episode.

As you recommended, we used our last remaining VST product sample from donor P0230D to perform a MLR with paternal bone marrow product cells to confirm there was no detectable alloreactivity in vitro, and if there was, to try and recover TCR sequences derived from alloreactive T cells. The frequency of proliferating T cells in the P0230D + donor BM cells mix was comparable to negative controls, and 5-fold lower than a positive control of P0230D cultured with an HLA mismatched donor. We then performed single-cell sorting on the small population of P0230D + donor BM cells that had appeared to have proliferated to some extent in the MLR and sent for TCR sequencing. However, we failed to retrieve any TCR sequences or amplification of beta-actin by RT-PCR, suggesting this was debris and not viable cells. In comparison, the viability of the non-proliferating cells was confirmed to be high via live/dead staining. This data has been added to Supplemental Figure 2.

In summary- the additional data obtained from these studies supports our assertion that VST expansion in this subject occurred due to extremely rare clones in the VST donor, and there was no detectable alloreactivity nor identifiable H-Y antigens in the product. These points have added this point to the discussion.

There are a number of problems with the data as presented:

1) The data in Supplemental Fig.2 are confusing for several reasons. One is that the stimulators are labelled as A02+ or A02-, but both the VST donor and the BM donor carry variants of HLA-A02 (A02:01 in the case of the BM donor and A02:11 for the VST donor). Presumably the stimulators do or do not express A02:01, but this needs to be indicated more clearly;

RESPONSE: We thank the reviewers for pointing this out and apologize for the confusion. We have added the donor HLA results to Supplemental Table 1. We re-performed the MLR experiments as described above, including with the bone marrow donor cells and, as control testing, with HLA-mismatched or autologous target cells. As shown in new Supplemental Figure 2, no VST proliferation in response to the bone marrow donor cells was observed.

2) The allogeneic stimulators in Supplemental Figure 2 presumably have extensive HLA mismatches from the VST in addition to HLA-A02. The entire HLA of the stimulators should be shown;

RESPONSE: As noted above, the mixed lymphocyte cultures were redone with HLA-mismatched irradiated PBMCs (HLA details added to Supplemental Table 1), irradiated cells from the bone marrow donor product, and autologous PBMCs.

3) It is stated in line 139 that pentamer staining with HLA-A02-restricted Y chromosome epitopes was negative, but this is not apparent from the cited figure, in which small percentages of positive cells are detected. A negative control stain is needed to determine whether these are real or not.

RESPONSE: We appreciate the reviewer's comments. We re-analyzed the flow staining data from that experiment and discovered that after cleaning the SSC/FSC gate to exclude debris, the small number of events in the pentamer+ gate cleared. We have updated Supplemental Figure 3 and included the tag-only control. In addition, we added the gating strategy details to Supplemental Figure 8, which we assert convincingly demonstrates that these stains are negative.

4) Line 177 states that T cells were sorted from the VST donor that reacted to Y antigens or the neoantigen pool, but the sorting method is not indicated. Was this done with tetramer/pentamer stains? The small pool of sorted positive cells seems likely to be heavily contaminated with nonspecifically stained cells. No methodology is provided for this sorting experiment.

RESPONSE: We apologize for this oversight and have added this to the methods.

5) Additional information on the TCR diversity of the VST would be useful.

RESPONSE: We thank the reviewer for this suggestion and have addressed this point above.

REVIEWERS' COMMENTS

Reviewer #1 (Remarks to the Author):

The authors have responded thoroughly to all my comments and I support the publication of this manuscript.

Reviewer #2 (Remarks to the Author):

My concerns have been addressed.